# The Impact of Two Recommended Withholding Periods for Omeprazole and the Use of a Nutraceutical Supplement on Recurrence of Equine Gastric Ulcer Syndrome in Thoroughbred Racehorses

**DOI:** 10.3390/ani13111823

**Published:** 2023-05-31

**Authors:** Ran Shan, Catherine M. Steel, Ben Sykes

**Affiliations:** 1Department of Veterinary Clinical Services, The Hong Kong Jockey Club, Conghua Racecourse, Guangzhou 510900, China; 2School of Veterinary Sciences, Massey University, Private Bag, 11-222, Palmerston North 4442, New Zealand

**Keywords:** equine gastric ulcer syndrome, omeprazole, gastric acid rebound

## Abstract

**Simple Summary:**

Ulceration of the stomach, also known as Equine Gastric Ulcer Syndrome (EGUS), is common in Thoroughbred racehorses. Ulceration can affect the upper (squamous) or lower (glandular) portion of the stomach with prevalence up to 100% and 25–72%, respectively, reported in racing populations. Omeprazole, a potent proton pump inhibitor drug, is widely used to treat this condition, with recommended withholding periods (RWPs) for racing varying from ‘not on race day’ to ‘2 clear days’. A rebound increase in gastric acid occurs in humans when acid-suppressing therapy with omeprazole ceases and might result in ‘rebound ulcers’. We hypothesized a similar phenomenon in horses if omeprazole treatment ceases in accordance with the RWPs for racing. We studied the effect of ‘not on race day’ (RWP0) and ‘2 clear days’ (RWP2) RWPs on the recurrence of EGUS in Thoroughbred horses in race training. Horses received a standard treatment course of omeprazole and were assessed by gastroscopic examination throughout the study. The prevalence of squamous ulceration was greater in horses subjected to an RWP2, although the effect was partially mitigated by the administration of a nutraceutical supplement, indicating potential implications of RWPs on the welfare and performance of Thoroughbred racehorses.

**Abstract:**

The impact of recommended withholding periods (RWPs) for omeprazole on the recurrence of Equine Gastric Ulcer Syndrome (EGUS) is unknown. The study was designed to compare the effect of two RWPs on EGUS recurrence post-omeprazole treatment and to determine if a nutraceutical supplement would reduce EGUS recurrence when administrated during an RWP. The study was a blinded, randomized clinical trial. Part 1: Horses were allocated to an RWP0 or RWP2 and crossed over after 4-weeks. Horses received oral omeprazole once daily, except during the RWPs at the end of the treatment periods. Part 2: Horses received omeprazole for 21 days prior to an RWP2 during which they received a nutraceutical supplement. Gastroscopy was performed on Day 0 and pre- and post- RWP. Part 1: More horses were affected by Equine Squamous Gastric Disease (ESGD) after the ‘2-clear-days’ RWP than the ‘not on race-day’ RWP (*p* = 0.012). The prevalence of ESGD post-RWP for ‘2-clear-days’ did not differ from day 0 (*p* = 0.478). Part 2: The prevalence of ESGD post-RWP was lower than on Day 0 (*p* = 0.046). A difference in recurrence of ESGD was present between the two common RWPs. The implications of this on the welfare of Thoroughbred racehorses warrant further discussion.

## 1. Introduction

The term Equine Gastric Ulcer Syndrome (EGUS) describes hyperemic, hyperkeratotic, erosive, and ulcerative disease of the stomach [1]. It is subdivided into Equine Squamous Gastric Disease (ESGD) and Equine Glandular Gastric Disease (EGGD) based on the region of the stomach that is affected [1]. Prevalence of up to 100% for ESGD [2] and 25–72% for EGGD [3,4,5,6,7] have been reported for Thoroughbred horses in race training. Numerous studies have documented the efficacy of short-term omeprazole therapy in the treatment and prevention of ESGD [8,9,10,11,12,13,14,15], but the rate of recurrence of the disease following discontinuation of treatment is poorly documented. Equine Squamous Gastric Disease has previously been reported to occur within 5 days of entering a simulated training environment [16] suggesting that ESGD can rapidly develop under ulcerogenic conditions, and both ESGD and EGGD recurred to pre-treatment prevalence within 28 days of discontinuation of omeprazole treatment [17]. Rebound gastric hyperacidity (RGH), which results when exaggerated production of hydrochloric acid occurs with the abrupt discontinuation of proton pump inhibitor (PPI) therapy [18], has been suggested as a potential contributor to ESGD recurrence [19]. The development of hypergastrinemia, considered to be the key contributor to RGH, has been demonstrated in horses after as little as 14 days of oral omeprazole at approximately 4 mg/kg once per day [20]. Whether RGH plays a role in the recurrence of EGUS following the discontinuation of omeprazole therapy has not been studied.

The recommended withholding period (RWP) for omeprazole varies between racing jurisdictions with RWP0 and RWP2 commonly used [21]. To the authors’ knowledge, the rate of recurrence of EGUS during these RWPs and following the discontinuation of omeprazole therapy has not been studied. However, observations by the authors within their clinical population (unpublished data) suggest that ESGD lesions can recur within as little as 3 days and within an RWP2. This suggests that EGUS can be present on race-day, despite treatment in the lead-up period. If present, such disease potentially could negatively impact the horse’s performance or welfare on race day [6,22,23,24].

The primary aim of this study was to compare the recurrence of ESGD and EGGD in Thoroughbred racehorses subjected to two different RWPs for omeprazole (RPW2 and RWP0). The secondary aim of the study was to evaluate the impact of feeding a commercial, nutraceutical supplement on the recurrence of ESGD and EGGD if given within the RWP2. It was hypothesized that horses subjected to RWP2 would have a higher prevalence of ESGD and EGGD on the mock race day than horses for which the RWP0 was applied and that the nutraceutical supplement would reduce the rate of recurrence of both ESGD and EGGD.

## 2. Materials and Methods

The study consisted of two parts, both blinded, clinical trials. It was approved by the Hong Kong Jockey Club Animal Ethics Committee (Permit number ERC/031/2021).

### 2.1. Materials of Part 1 Study

#### 2.1.1. Animals

Retired racehorses used for jockey training were sourced from the Hong Kong Jockey Club’s Conghua Racecourse Jockey Training School. Horses were included if they were normal on clinical and lameness examination, and the trainer considered the horse to be suitable for a conditioning program over a 10-week period to conduct a mock race on 2 separate occasions. Horses were excluded if they had received omeprazole, or another acid-suppressive drug, in the two months prior to enrollment in Part 1 of the study. Throughout the study, animals remained in their usual stable environment and were fed their usual diet of timothy hay (7 kg each morning) and lucerne hay (5 kg each afternoon), commercial complete feeds (1.8 kg Connolly’s Red Mills 10% (Connolly’s Red Mills, Goresbridge, Ireland) and 0.45 kg of 14% Connolly’s Red Mills 14% (Connolly’s Red Mills, Goresbridge, Ireland) twice daily, 0.18 kg fermented lucerne (FiberProtect™, FiberFresh, Reporoa, New Zealand) each morning, 1.3 kg rice bran (EquiJewel™, Kentucky Equine Research, Versailles, KY, USA) twice daily, and 70 g corn oil twice daily. Horses were not fed pre-exercise on the days that the gastroscopy was performed. All horses underwent a similar exercise training program in the same stable under the same trainer throughout the study period.

A power calculation was performed using an online calculator (https://select-statistics.co.uk/calculators/sample-sizecalculator-two-proportions/ (accessed on 9 May 2023)) based on the following assumptions; it was estimated that 75% of horses in the ‘2-clear-day’ group would develop gastric lesions compared with 25% of horses in the ‘Not on race day’ group. Assuming a confidence level of 95% and a power of 80% a sample size of 12 horses per group was determined. To allow for a drop-out rate of 2 horses per group due to injury or illness, 14 horses were recruited with each horse serving as its own control.

#### 2.1.2. Part 1—Drug Administration

From Day 1 to Day 61 (except for withholding periods), horses received 2.4 g (4.4–5.3 mg/kg for approximately 450–550 kg horses) of a commercially available omeprazole formulation (GastroGard™, Boehringer Ingelheim, Ridgefield, CT, USA) daily by mouth at 4 AM each morning. To maximize absorption and activation, the omeprazole was administered 30 min before morning feeding with 0.5 kg of a commercial fiber supplement [25]. Feeding was performed pre-exercise both during treatment and during the RWPs.

#### 2.1.3. Part 1—Recommended Withholding Periods

Part 1 of the study compared two RWPs (‘2-clear-day’ vs. ‘Not on race day’). Fourteen horses were allocated into two equal groups at enrollment. Allocation was done by the trainer, who was not informed of the purpose of the splitting, based on the requirement for having two similar groups. Group A was subjected to the RWP2 withhold at the end of week 4 of treatment (withheld for days 29–31 inclusive) and to the RWP0 withhold at the end of week 8 of treatment (withheld for day 62 only). Group B was subjected to the RWP0 withhold at the end of week 4 of treatment (withheld for day 31 only) and to the RWP2 withhold at the end of week 8 of treatment (withheld for days 60–62 inclusive).

#### 2.1.4. Part 1—Gastroscopy

Gastroscopy was performed on Days 0, 28 (pre-withhold), 31 (post-withhold), 59 (pre-withhold), and 62 (post-withhold), the time point of each gastroscopy examination was illustrated in Figure 1. Gastroscopy was performed after morning training or 1–2 h after each ‘mock race’ where applicable. For gastroscopy, horses were sedated intravenously with detomidine (Dozadine™, Virbac, Milperra, Australia; 0.02 mg/kg BW IV) or xylazine hydrochloride (Xylazil-100™, Troy Laboratories, Glendenning, Australia; 0.5 mg/kg BW IV) with or without butorphanol (Torbugesic™, Zoetis, Surrey, UK; 0.02 mg/kg BW IV) and were examined for the presence of EGUS using a portable 3-m-long video gastroscope (60130PKS, Karl Storz, Tuttlingen, Germany; Colonix 98V(HD), Shinova, Shanghai, China). Videos and still images of the stomach were recorded for each examination. One of the investigators (B.W. Sykes), who was blinded to the study group and day of examination, evaluated and graded ESGD and EGGD gastric lesions using a 0–4/4 scale [26]. Lesions were considered clinically significant if they were graded ≥2/4 [3,4,5,6,27].

### 2.2. Materials of Part 2 Study

#### 2.2.1. Part 2—Animals

A wash-out period of 9 weeks was allowed between Part 1 and Part 2 of the study. Horses from the same stable as in Part 1 were recruited for Part 2 of the study using the same inclusion criteria as Part 1. Horses were housed, fed, and exercised as described in Part 1 of the study.

#### 2.2.2. Part 2—Recommended Withholding Period, Drug and Nutraceutical Administration

All horses were subject to the RWP2 in Part 2 of the study. All horses were given 2.4 g of oral omeprazole daily (as described in Part 1) for 21 days. On Days 22 and 23 (i.e., during the 2-day withholding period but not on ‘race’ day) horses received 250 g (divided evenly into two meals—morning and evening) of a commercially available nutraceutical supplement (GastroAid Recovery™; Kelato, Miranda, NSW, Australia) [17]. The morning meal that contained the nutraceutical supplement was fed pre-exercise with 0.18 kg fermented lucerne (FiberProtect™, FiberFresh, Reporoa, New Zealand). In accordance with the Hong Kong Jockey Club racing rules that prohibit the use of a supplement with alkalinizing effects, horses did not receive nutraceutical supplementation on the day of the ‘mock race’. To facilitate gastroscopy, horses were not fed pre-exercise on the day of the ‘mock race’.

#### 2.2.3. Part 2—Gastroscopy

Gastroscopy was performed on Days 0, 21 (pre-withhold), and 24 (post-withhold). Gastroscopy was performed after morning training or 1–2 h after the ‘mock race’. Images were recorded and lesions were graded as described in Part 1 of the study.

### 2.3. Statistical Analysis

Data was collated into an Excel spreadsheet. Data are presented as % affected with 95% confidence intervals. Confidence intervals were calculated using Jeffreys’ intervals and an online calculator (https://epitools.ausvet.com.au/ciproportion (accessed on 9 May 2023)). A Fishers Exact Test was used for all comparisons using an online calculator (https://www.socscistatistics.com/tests/fisher/ (accessed on 9 May 2023)). Significance was defined as *p* ≤ 0.05.

For Part 1 of the study: The percentage of horses affected with ESGD and EGGD ≥ grade 2/4 pre-withhold (Days 28 and 59) for both RWPs were compared with baseline values (Day 0). No effect of treatment RWP was seen on EGGD, and as such no further statistical analysis was performed. The percentage of horses affected with ESGD ≥ grade 2/4 was compared between pre-withhold (Days 28 and 59) and post-withhold (Days 31 and 62) for both RWPs. The percentage of horses affected with ESGD post-withhold (Days 31 and 62) were compared between the ‘Not on race day’ and ‘2 clear days’ RWPs. The percentage of horses affected with ESGD ≥ grade 2/4 was compared between baseline (Day 0) and the post-withholding (Days 31 and 62) values for both RWPs.

For Part 2 of the study: The percentage of horses affected with ESGD ≥ grade 2/4 pre-withhold (Day 21) were compared with baseline values (Day 0). The percentage of horses affected with ESGD ≥ grade 2/4 was compared between pre-withhold (Day 21) and post-withhold (Day 24) values. The percentage of horses affected with ESGD ≥ grade 2/4 was compared between baseline (Day 0) and the post-withholding (Day 24) values.

## 3. Results

### 3.1. Animals

Fourteen Thoroughbred geldings aged from 8 to 14 years old were enrolled in Part 1 of the study, all horses were actively in training at enrolment. Two were subsequently withdrawn when training ceased due to unrelated reasons (one due to musculoskeletal injury and one transferred to another stable). These horses were excluded from the analysis. Fourteen Thoroughbred geldings aged from 8 to 14 were used in Part 2 of the study, all horses were actively in training at enrolment and all successfully completed the study. Seven of the 14 horses in Part 2 participated in Part 1 of the study. Omeprazole and the nutraceutical supplement were administered according to the protocol in all horses and all RWPs were observed as planned.

### 3.2. Gastroscopy

#### 3.2.1. Enrolment

In Part 1, all horses had ESGD (12/12; 100%; 95% CI 81–100%) and 7/12 (58%; 95% CI 31–82%) had EGGD on Day 0. In Part 2 of the study, 12/14 (86%; 95% CI 62–97%) had ESGD and 8/14 (57%; 95% CI 32–80%) had EGGD on Day 0.

#### 3.2.2. Part 1—Equine Squamous Gastric Disease

When compared with Day 0 (12/12; 100%; 95% CI 81–100%), treatment with omeprazole resulted in a reduction in the prevalence of ESGD on pre-withhold Day 28 (4/12; 33%; 95% CI 12–61%; *p* = 0.001) and 59 (3/12; 25%; 95% CI 8–53%; *p* < 0.001).

No effect was observed for the RWP0 with the percentage of horses affected not different between the pre-withhold (4/12; 33%; 95% CI 12–61%) and post-withhold (3/12; 25%; 95% CI 8–53%) examinations (*p* = 1.00). In contrast, worsening was observed during the RWP2 with a higher percentage of horses affected post-withhold (10/12; 83%; 95% CI 56–96%) than pre-withhold (2/12; 17%; 95% CI 4–44%; *p* = 0.003). A greater prevalence of ESGD was present after the RWP2 (10/12; 83%; 95% CI 56–96%) than the RWP0 (3/12; 25%; 95% CI 8–53%; *p* = 0.012).

The prevalence post-withhold (3/12; 25%; 95% CI 8–53%) was lower than Day 0 values (12/12; 100%; 95% CI 81–100%) for the RWP0 (*p* < 0.001). In contrast, there was no difference between post-withhold prevalence (10/12; 83%; 95% CI 56–96%) and Day 0 prevalence (12/12; 100%; 95% CI 81–100%) for the RWP2 (*p* = 0.478). The data over time for ESGD for the two RWPs are summarized in Figure 2.

#### 3.2.3. Part 1—Equine Glandular Gastric Disease

When compared with Day 0 (7/12; 58%; 95% CI 31–82%), no effect of omeprazole treatment was observed on pre-withhold Days 28 or 59 (3/12; 25%; 95% CI 8–53% for both time points; *p* = 0.214). The data over time for EGGD for the two RWPs are summarized in Figure 3.

#### 3.2.4. Part 2—Equine Squamous Gastric Disease

When compared with Day 0 (12/14; 86%; 95% CI 62–97%) treatment with omeprazole resulted in a reduction in the prevalence of ESGD on pre-withhold Day 21 (2/14; 14%; 95% CI 3–38%; *p* < 0.001). There was no effect of the RWP on the prevalence of ESGD (6/14; 43%; 95% CI 20–68% post-withhold vs. 2/14; 14%; 95% CI 3–38% pre-withhold; *p* = 0.209). The prevalence of ESGD post-withhold (6/14; 43% 95% CI 20–68%) was lower than on Day 0 (12/14; 86%; 95% CI 62–97%; *p* = 0.046). The data over time for ESGD for Part 2 of the study is summarized in Figure 4.

## 4. Discussion

The primary aim of this study was to compare the recurrence of ESGD and EGGD in Thoroughbred racehorses subjected to two RWPs for omeprazole (‘2 clear days’ and ‘Not on race day’). The hypothesis that horses subjected to an RWP2 would have a higher prevalence of ESGD than horses subjected to an RWP0 was supported. Further, the study demonstrated that the prevalence of ESGD can return to pre-treatment levels in as little as 3 days. No effect of treatment RWP was seen on EGGD and as such further analysis was not conducted. The secondary aim of the study was to evaluate the impact of feeding a commercial, nutraceutical supplement on the recurrence of ESGD if given within a RWP2. The hypothesis that the nutraceutical supplement would reduce the rate of recurrence of ESGD was, within the limitations of the study design, also supported as the recurrence rate of ESGD was reduced from 83% in the Part 1 study to 43% in the Part 2 study. As no effect of an RWP was observed on EGGD, the hypothesis that the nutraceutical supplement would reduce the rate of recurrence of EGGD could not be tested.

The high prevalence of ESGD (84–100%) and EGGD (57–58%) at enrollment are consistent with previous reports for Thoroughbred racehorses [2,3,4,5,7]. This finding is not unexpected as key contributory factors to ESGD such as high carbohydrate/low roughage diets and high volume/high-intensity exercise are part of the routine management of the study population. Similarly, although the risk factors for EGGD are less well described, the prevalence reported here is comparable to that previously reported for Australian Thoroughbred racehorses managed under similar conditions [3,4,5]. Similarly, the positive response of ESGD to omeprazole is consistent with previous reports [1], as is the ineffectiveness of monotherapy with omeprazole for the treatment of EGGD in Thoroughbred racehorse populations [3,4,5].

The rate of recurrence of ESGD within the RWP2 is faster than previously described, where horses in a simulated show horse training environment were documented to develop ESGD within 5 days of starting training. The authors propose two key mechanisms that likely contributed to this effect. Firstly, as above, the training and management of the environment, which is typical of high-performance Thoroughbred racehorses, is well documented to be highly ulcerogenic [1]. Secondly, the authors propose that RGH might play a role in the rapid recurrence of the disease. Rebound gastric hyperacidity is a well-recognized problem of abrupt discontinuation of omeprazole therapy in humans with gastric acid production rising above pre-treatment levels in affected individuals [28]. Up to 44% of healthy human volunteers receiving a 4-week course of acid suppressor therapy experienced symptoms consistent with RGH, such as severe heartburn, in one study [29]. In another study, approximately 22% of healthy human volunteers experienced symptoms consistent with RGH after 8 weeks of acid suppression therapy [30]. The best-established mechanism by which RGH occurs in humans is through elevations in serum gastrin concentrations [18]. Gastrin is produced by the epithelial G-cells of the pyloric antrum, pancreas, and duodenum in response to a range of stimuli associated with feeding [18]. Conversely, its production is inhibited in a negative feedback loop by decreased intra-gastric pH via somatostatin release from D-cells [18]. Serum gastrin stimulates intra-gastric acid secretion directly by stimulation of the parietal cell, and indirectly through ECL-cell activation, which, in turn, causes the release of histamine, another potent stimulator of the parietal cell [31]. The ECL-cell/histamine pathway is the dominant signaling mechanism for gastric acid production [18]. Acid suppressive drugs increase intra-gastric pH and result in loss of this negative feedback on gastrin production that subsequently causes hypergastrinemia [18]. In addition to direct stimulatory effects on gastric acid production, gastrin also has trophic effects on a range of gastric mucosa cells, especially the ECL-cell with prolonged hypergastrinemia resulting in proliferation of the ECL-cell population [32]. When acid suppressive therapy is discontinued, RGH is observed in response to hypergastrinemia. It has recently been demonstrated that oral omeprazole therapy induces hypergastrinemia in the horse with serum gastrin levels doubling over a 14-day period with omeprazole administered at approximately 4 mg/kg BW/day [20]. Whether hypergastrinemia and RGH contributed to the rapid rate of recurrence observed in the present study is unclear but warrants further investigation.

The use of a commercial nutraceutical supplement during the RWP appeared to reduce the rate of recurrence of ESGD in the present study. This is consistent with a previous study that evaluated the effect of a near-identical formulation over a 28-day period following the discontinuation of omeprazole treatment, in which protective effects for both ESGD and EGGD were demonstrated [17]. However, an important limitation of the present study was that no control group was used in Part 2 of the study, with the results from Part 1 being used as the control instead. Although the horses were otherwise managed under identical conditions, the different durations of treatment (21 days in Part 2 vs. 28 days in Part 1) might have influenced the likelihood of RGH as a contributory factor. The authors consider this unlikely as the magnitude of hypergastrinemia, the main driver for RGH does not differ between 3 and 4 weeks of omeprazole treatment (unpublished data). Given these limitations, further study using an appropriate control group is required before firm conclusions can be drawn. Other nutraceuticals have been demonstrated to have protective effects against ESGD under similar conditions [33]. As such, the strategic use of specific nutraceuticals during RWPs or following the discontinuation of omeprazole might be useful in reducing the risk of recurrence of ESGD.

A further limitation of the present study is the small sample size. Although an effect of RWP was able to be demonstrated for ESGD, no such effect was observed for EGGD. The raw data in Part 1 of the study suggested that EGGD may be more common following the RWP2 than the RWP0. Whether such an effect is present warrants further investigation in a larger population, however in the interim no conclusions should be drawn in this regard. A final potential limitation of the present study is that horses were not fed on the morning before the mock race gallop as this exercise was conducted in the morning and gastroscopic examination was performed within several hours of the exercise. This might have increased the prevalence of ESGD at repeat examinations.

## 5. Conclusions

In conclusion, the present study demonstrated a difference in the rate of recurrence between two common RWPs. The implications of this finding on the welfare and performance of Thoroughbred racehorses and the balance between those needs, and the need for the integrity of racing to be preserved warrants further discussion. Further, the present study demonstrated that the use of a commercial, nutraceutical supplement mitigated the rate of recurrence of ESGD in a high-risk setting.

## Figures and Tables

**Figure 1 animals-13-01823-f001:**
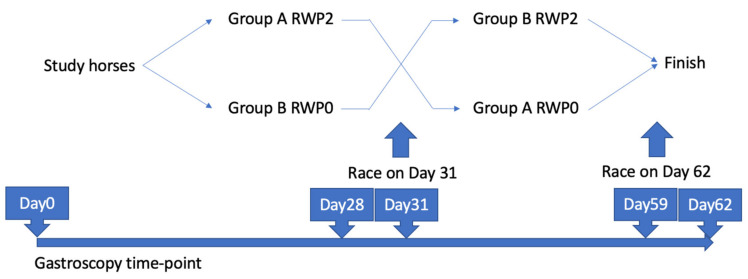
All horses were divided into two groups; group A was subjected to RWP2 and group B was subject to RWP0, crossed over after Day 31 (first mock race day). Gastroscopy examination was performed on Days 0, 28, 31, 59, and 62.

**Figure 2 animals-13-01823-f002:**
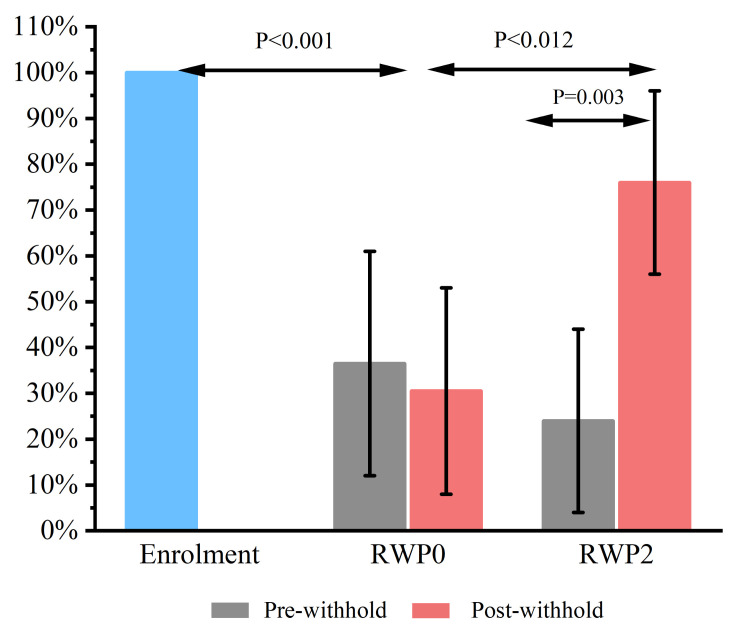
The impact of two Recommended Withholding Periods (RWPs) for omeprazole (‘RWP0′ vs. ‘RWP2’) on recurrence of Equine Squamous Gastric Disease (ESGD) in 12 Thoroughbred racehorses. The y-axis represents the percentage of horses affected by ESGD. Differences between the groups are noted with corresponding *p* values.

**Figure 3 animals-13-01823-f003:**
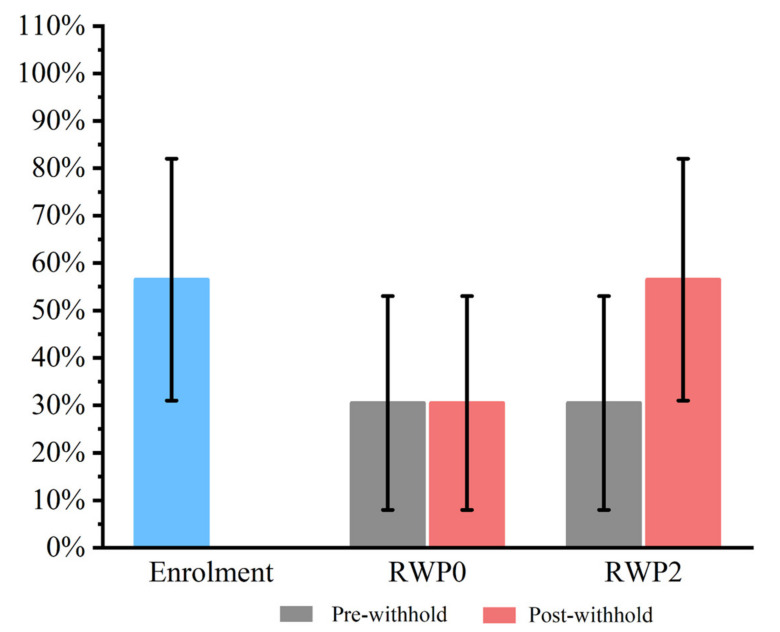
The impact of two Recommended Withholding Periods (RWPs) for omeprazole (‘RWP0′ vs. ‘RWP2’) on recurrence of Equine Glandular Gastric Disease (EGGD) in 14 Thoroughbred racehorses. The y-axis represents the percentage of horses affected by EGGD. No differences were observed between the groups.

**Figure 4 animals-13-01823-f004:**
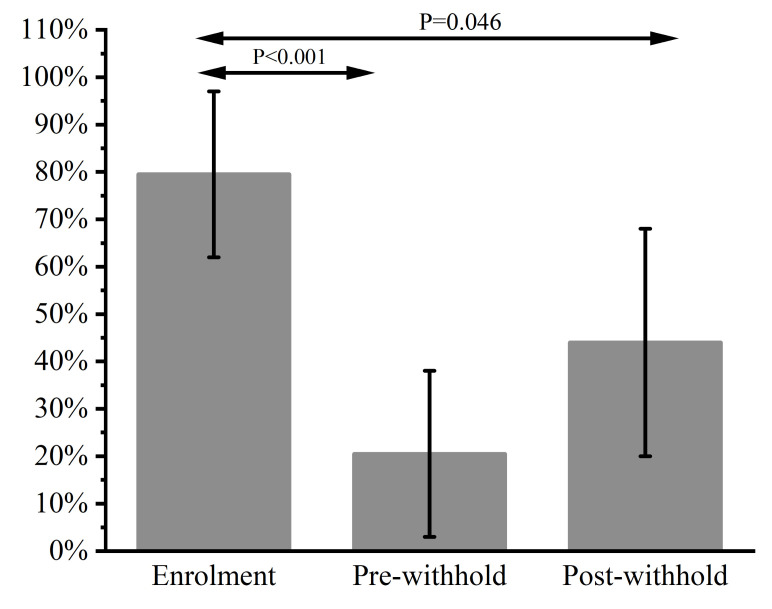
The impact of a commercial, nutraceutical supplement on the recurrence of Equine Squamous Gastric Disease (ESGD) during an ‘RWP2′ Recommended Withholding Period (RWP) for omeprazole in 14 Thoroughbred racehorses. The y-axis represents the percentage of horses affected by ESGD.

## Data Availability

Not applicable.

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
