# Peer review of "The Impact of Two Recommended Withholding Periods for Omeprazole and the Use of a Nutraceutical Supplement on Recurrence of Equine Gastric Ulcer Syndrome in Thoroughbred Racehorses"

_animals, 2023, doi:10.3390/ani13111823_

Round 1

Reviewer 1 Report

I'm not sure that the conclusions about part 2 (line 311 The use of a commercial nutraceutical supplement during the RWP reduced the rate  of recurrence of ESGD in the present study.) are supported by the results of this part of the experiment.  There was no control group in this part of the study so it is not clear what rate of recurrence the results were compared to.  The statistical analysis of this part of the study only compared the incidence of ulcers to other time points within the same study.  Are the authors comparing the day 24 recurrence incidence to the results of part 1?  if so, that should be clearly stated.

Author Response

The conclusion of this effect was drawn from comparing the recurrence rate of ESGD of the ‘2 clear day’ groups between Study 1 and 2.  The authors are aware of the limitation of the lack of a control group, and this has been stated in the resubmitted manuscript to draw the reader’s attention to the same.  The wording has also been adjusted to soften the importance of the finding.

Reviewer 2 Report

The title is missing information on nutraceutical use. I assume that the authors wanted to keep it brief but I suggest a change to incorporate that and also add the term nutraceutical to the keywords, as otherwise this information will be lost on potential readers while looking for information on these subjects.  

Please recheck the paper for minor text editing (for example row 134 should read Karl Storz endoscopes, not Karl Stroz, or row 271, where the line reads high volume/high-intensity exercise1- I am unsure whether that is a typo or more probably a missed reference). Also, according to "Animals" instructions for authors, in the text of the paper, reference numbers should be placed in square brackets and placed BEFORE punctuation, for example  [1], [1–3], or [1,3]. 

Author Response

The title is missing information on nutraceutical use. I assume that the authors wanted to keep it brief, but I suggest a change to incorporate that and also add the term nutraceutical to the keywords, as otherwise this information will be lost on potential readers while looking for information on these subjects.  

Changed as suggested.

Please recheck the paper for minor text editing (for example row 134 should read Karl Storz endoscopes, not Karl Stroz, or row 271, where the line reads high volume/high-intensity exercise1- I am unsure whether that is a typo or more probably a missed reference). Also, according to "Animals" instructions for authors, in the text of the paper, reference numbers should be placed in square brackets and placed BEFORE punctuation, for example  [1], [1–3], or [1,3]

Addressed as suggested.

Reviewer 3 Report

Abstract and summary -- adequate

Introduction - adequate,

Line 14 -- remove one of the "glandular" designations

section 3.2.3  -- line 218-219.  I am a bit confused by the reporting of the data analysis -- in these sentences for example you report the number of horses with ulcers day 0 vs  pre withhold -- 7 of 12 had ulcers day 0, 3/12 had ulcers pre-witholding and you report a p value for this comparison of .414  -- this seemed surprising to me and i did the chi-square analysis myself ( 7 yes day 0, 5 no day o, 3 yes pre, 9 no pre) and got a p value of .088. Clearly i am not understanding your data analysis structure - can you clarify please. 

Would recommend to report Odds ratios also as a measure of effect size

there are a large number of comparisons conducted in this study (perhaps too many-- there are only a very few of these that are specific to the hypothesis being tested) -- these are adequately explained in Section 2.3 but it might improve the manuscript to lay these out in a timeline, showing the days of examination, and using some markers or bars (as in the current figures for example) to show the various comparisons.

Your study design is pretty complex, if you choose to analyze the data using only binomial comparisons, i would suggest you might consider reducing some of the comparisons that are not particularly germaine to the hypothesis being tested. Otherwise, a more robust method of analysis which accounts for multiple pairways analyses and paired subjects should be considered. 

Similarly, in line 283, for example you report values of 6 /14 post withhold  and 2/14 pre withhold with a p value of .70 reported-- when i calculate this i get a p value of .088 -- again -- am i misunderstanding your data analysis?  

Line 258 -- i would say that your hypothesis was supported, "proven" is i think a bit too strong of a word

Line 255 -- true- that specific question cannot be addressed, but you can (should)  comment about the effectiveness of the nutraceutical itself given that horses with ulcers were administered the drug and an effect was evaluated.

Can the authors comment upon the potential effects of race-day feed or dietary changes that might have influenced the results ?

Figures and legends- well presented

References -- adequate.

Author Response

  1. Line 14 -- remove one of the "glandular" designations

Changed as suggested.

  1. section 3.2.3 -- line 218-219.  I am a bit confused by the reporting of the data analysis -- in these sentences for example you report the number of horses with ulcers day 0 vs  pre withhold -- 7 of 12 had ulcers day 0, 3/12 had ulcers pre-witholding and you report a p value for this comparison of .414  -- this seemed surprising to me and i did the chi-square analysis myself ( 7 yes day 0, 5 no day o, 3 yes pre, 9 no pre) and got a p value of .088. Clearly i am not understanding your data analysis structure - can you clarify please. 

Thank you for picking up these errors in transcription. The analysis of each comparison has been checked using a different calculator (graph pad online) and by an independent person with two corrections made (the ones you highlighted).  Please note that a Fisher’s Exact Test has been used, not Chi-Squared as Fisher’s Exact Test is more appropriate for smaller data sets such as this.  This explains the differences in the final p values reported and the ones you calculated. 

  1. Would recommend reporting Odds ratios also as a measure of effect size

As per the recommendations below we have reduced the number of comparisons and simplified the data analysis.  We acknowledge the potential benefit of odds ratios but believe that in this circumstance that to add them would be counterproductive to that overall goal of simplified analysis as below.

  1. there are a large number of comparisons conducted in this study (perhaps too many-- there are only a very few of these that are specific to the hypothesis being tested) -- these are adequately explained in Section 2.3 but it might improve the manuscript to lay these out in a timeline, showing the days of examination, and using some markers or bars (as in the current figures for example) to show the various comparisons.

As suggested – the number of statistical comparisons has been reduced with the EGGD removed beyond the initial assessment where no treatment effect was seen (meaning that it is technically not possible to assess recurrence). Similarly, and for the same reasons, analysis of EGGD data has been removed from part 2. This has effectively halved the number of statistical comparisons.  

  1. Your study design is pretty complex, if you choose to analyze the data using only binomial comparisons, i would suggest you might consider reducing some of the comparisons that are not particularly germaine to the hypothesis being tested. Otherwise, a more robust method of analysis which accounts for multiple pairways analyses and paired subjects should be considered. 

As suggested – the number of statistical comparisons has been reduced with the EGGD removed beyond the initial assessment where no treatment effect was seen (meaning that it is technically not possible to assess recurrence).  Similarly, and for the same reasons, analysis of EGGD data has been removed from part 2.  This has effectively halved the number of statistical comparisons reducing the risk of a type I error from excessive multiple comparisons.

  1. Similarly, in line 283, for example you report values of 6 /14 post withhold and 2/14 pre withhold with a p value of .70 reported-- when i calculate this i get a p value of .088 -- again -- am i misunderstanding your data analysis?  

As above - Thank you for picking up these errors in transcription.  The analysis of each comparison has been checked using a different calculator (graph pad online) and by an independent person with two corrections made.  Please note that a Fisher’s Exact Test has been used, not Chi-Squared as Fisher’s Exact Test is more appropriate for smaller data sets such as this.  This explains the differences in the final p values reported and the ones you calculated. 

  1. Line 258 -- i would say that your hypothesis was supported, "proven" is i think a bit too strong of a word

Changed as suggested.

  1. Line 255 -- true- that specific question cannot be addressed, but you can (should) comment about the effectiveness of the nutraceutical itself given that horses with ulcers were administered the drug and an effect was evaluated.

As per addressing the comment from reviewer 1 - The conclusion of this effect was drawn from comparing the recurrence rate of ESGD of the ‘2 clear day’ groups between Study 1 and 2.  The authors are aware of the limitation of the lack of a control group, and this has been stated in the resubmitted manuscript to draw the reader’s attention to the same.  The wording has also been adjusted to soften the importance of the finding.

  1. Can the authors comment upon the potential effects of race-day feed or dietary changes that might have influenced the results?

All horses did not receive feed on the race-day in this study to facilitate the post-race gastroscopy examination. It will be hard to compare between horses with feeding and without feeding since horses should be fastened for gastroscopic examination. A comment has been added to the discussion to acknowledge this.

Round 2

Reviewer 1 Report

Modifications to manuscript are satisfactory.

Author Response

Thanks for your comments, we have modified the manuscripts per request.

Reviewer 3 Report

No further comments

Author Response

(The authors gave the same response as above.)
